# PhoPQ-mediated lipopolysaccharide modification governs intrinsic resistance to tetracycline and glycylcycline antibiotics in *Escherichia coli*

Byoung Jun Choi,[1] Umji Choi,[1] Dae-Beom Ryu,[1] Chang-Ro Lee[1]

**ABSTRACT**    Tetracyclines and glycylcycline are among the important antibiotics used to combat infections caused by multidrug-resistant Gram-negative pathogens. Despite the clinical importance of these antibiotics, their mechanisms of resistance remain unclear. In this study, we elucidated a novel mechanism of resistance to tetracycline and glycylcycline antibiotics via lipopolysaccharide (LPS) modification. Disruption of the *Escherichia coli* PhoPQ two-component system, which regulates the transcription of various genes involved in magnesium transport and LPS modification, leads to increased susceptibility to tetracycline, minocycline, doxycycline, and tigecycline. These phenotypes are caused by enhanced expression of phosphoethanolamine transferase EptB, which catalyzes the modification of the inner core sugar of LPS. PhoPQ-mediated regulation of EptB expression appears to affect the intracellular transportation of doxycycline. Disruption of EptB increases resistance to tetracycline and glycylcycline antibiotics, whereas the other two phosphoethanolamine transferases, EptA and EptC, that participate in the modification of other LPS residues, are not associated with resistance to tetracyclines and glycylcycline. Overall, our results demonstrated that PhoPQ-mediated modification of a specific residue of LPS by phosphoethanolamine transferase EptB governs intrinsic resistance to tetracycline and glycylcycline antibiotics.

**IMPORTANCE** Elucidating the resistance mechanisms of clinically important antibiotics helps in maintaining the clinical efficacy of antibiotics and in the prescription of adequate antibiotic therapy. Although tetracycline and glycylcycline antibiotics are clinically important in combating multidrug-resistant Gram-negative bacterial infections, their mechanisms of resistance are not fully understood. Our research demonstrates that the *E. coli* PhoPQ two-component system affects resistance to tetracycline and glycylcycline antibiotics by controlling the expression of phosphoethanolamine transferase EptB, which catalyzes the modification of the inner core residue of lipopolysaccharide (LPS). Therefore, our findings highlight a novel resistance mechanism to tetracycline and glycylcycline antibiotics and the physiological significance of LPS core modification in *E. coli*.

**KEYWORDS** lipopolysaccharide modification, antibiotic resistance, tetracycline, glycylcycline, tigecycline, minocycline, doxycycline, PhoPQ

In a single cell, bacteria must be equipped with all the stress response mechanisms necessary to overcome diverse environmental stresses. Therefore, bacteria have precise stress adaptation mechanisms (1), such as the two-component system. The two-component system comprises of a sensor kinase and response regulator (2). Sensor kinase senses extracellular stress conditions and induces autophosphorylation at a specific residue of its cytoplasmic domain (2). Subsequently, the phosphate group of the

Address correspondence to Chang-Ro Lee, crlee@mju.ac.kr.

Byoung Jun Choi and Umji Choi contributed equally to this article. Author order was determined both alphabetically and in order of increasing seniority.

The authors declare no conflict of interest.

See the funding table on p. 13.

sensor kinase is transferred to a specific residue of the response regulator. Phosphorylated response regulators can induce or repress the transcription of various target genes (2).

The PhoPQ system is an important two-component system involved in adaptation to envelope stresses. The sensor kinase PhoQ phosphorylates the response regulator PhoP in response to various conditions, such as magnesium depletion (3), acidic stress (4), exposure to cationic antimicrobial peptides (5), and osmotic upshift (6). Phosphorylated PhoP regulates the transcription of several genes, such as the *mgtA* gene encoding a magnesium transporter that mediates the import of magnesium under magnesium starvation conditions (7, 8). The PhoPQ system also plays a pivotal role in the survival of intracellular bacterial pathogens such as *Salmonella enterica*, inside macrophage (9, 10). Although the PhoPQ system enables bacteria to survive under diverse environmental conditions, the regulatory mechanisms underlying PhoPQ-mediated antibiotic resistance remain unclear.

Tetracycline and its derivatives such as minocycline and doxycycline are used to treat infections caused by multidrug-resistant Gram-negative pathogens (11). Tigecycline is a unique antibiotic in the glycylcycline class derived from tetracycline, and tigecycline is among the last-resort antibiotics for the treatment of severe infections caused by multidrug-resistant Gram-negative pathogens (12, 13). Owing to their clinical importance, the resistance mechanisms to tetracyclines and glycylcycline, such as efflux pumps, ribosome modifications, and the production of inactivation enzymes, have been investigated in several Gram-negative pathogens (11, 14–16). However, the impact of altered outer membrane permeability, such as lipopolysaccharide (LPS) modification, on the resistance to tetracycline and glycylcycline antibiotics is not fully understood.

In this study, we demonstrated that LPS core modification governs resistance to tetracycline and glycylcycline antibiotics in *Escherichia coli*. The PhoPQ system represses the expression of phosphoethanolamine transferase EptB, which catalyzes the modification of the inner core sugar of LPS, which is necessary for resistance to tetracycline and glycylcycline antibiotics. Additionally, of the three phosphoethanolamine transferases (EptA, EptB, and EptC) involved in LPS modification, only EptB is associated with resistance to tetracyclines and glycylcycline. Overall, these results suggest that PhoPQ-mediated regulation of EptB expression affects resistance to tetracycline and glycylcycline antibiotics.

## RESULTS

### Inactivation of sensor kinase PhoQ induces increased susceptibility to minocycline and tigecycline

We constructed several mutants defective in sensor kinases associated with the envelope stress response to determine the effect of the two-component system on antibiotic resistance, and examined the minimal inhibitory concentrations (MICs) of 32 antibiotics with different modes of action, such as β-lactams, aminoglycosides, tetracyclines, quinolones, anti-folates, glycylcycline, glycopeptide, macrolide, lincosamide, amphenicol, aminocoumarin, and other antibiotics, in these mutant strains. All mutant strains did not show any growth defect in LB medium, except for slight growth retardation of the ΔcpxA mutant (Fig. S1). Among the 11 mutant strains, 6 mutant strains did not exhibit any change in the MICs of the antibiotics tested (Fig. S2), whereas 5 mutant strains exhibited changes in the MICs for one or more antibiotics (Fig. 1). The MICs of β-lactams (blue bars), aminoglycosides (orange bars), and fosfomycin (yellow bar) were increased in the ΔcpxA mutant, compared to those in the wild-type strain (Fig. 1). A previous study reported that CpxA affects resistance to β-lactams via regulation of *ompF* and *slt* expressions, aminoglycosides resistance via regulation of *acrD* expression, and fosfomycin resistance via regulation of *glpT* and *uhpT* expressions (17). β-lactam resistance was also affected in the ΔenvZ mutant (green bars) (Fig. 1). These results may be owing to the altered expression levels of OmpF and OmpC porins, which strongly affect β-lactam resistance through the penetration of the outer membrane and the maintenance of membrane

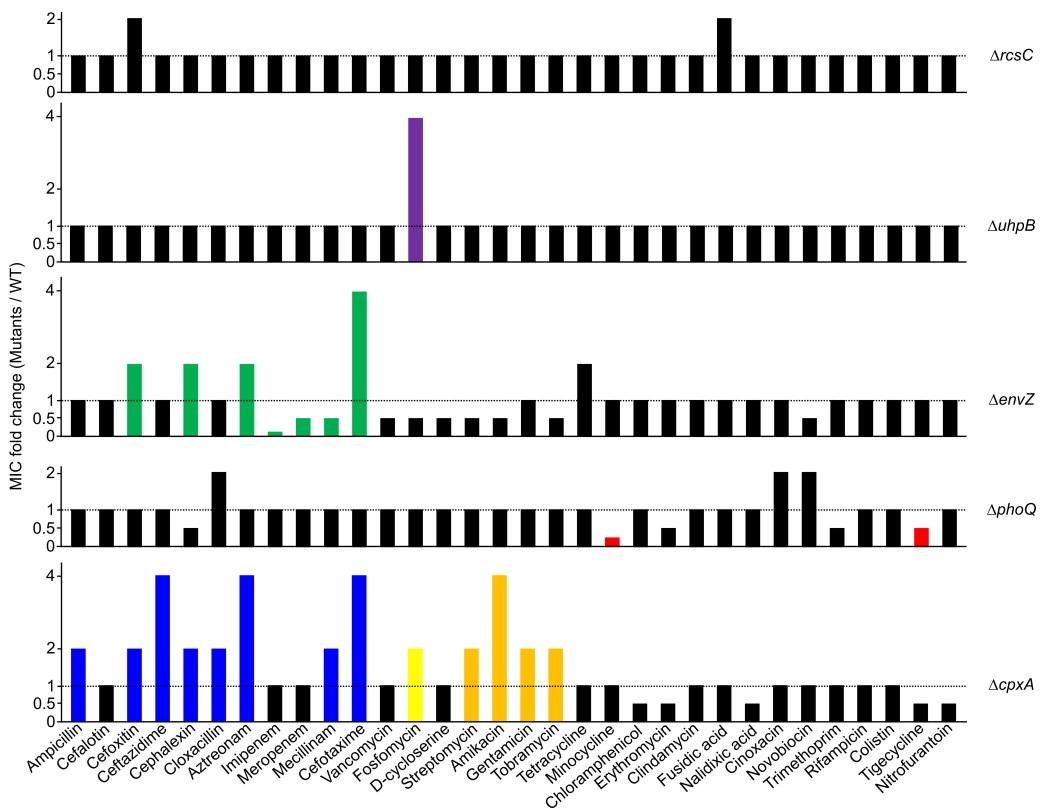

**FIG 1** The inactivation of two-component system sensor kinases affects intrinsic antibiotic resistance. The MICs of various antibiotics were measured against the wild-type and indicated mutant strains in MH medium. The relative MIC values for the indicated mutant cells compared to those for the wild-type cells are presented. Colored bars indicate the MIC values of important antibiotics, which increase or decrease in the indicated mutant cells.

integrity (18). The MIC of fosfomycin was 4-fold higher in Δ*uhpB* mutant than in the wild-type strain (purple bar) (Fig. 1). Previous studies have revealed that UhpB affects fosfomycin resistance via regulation of the expression of a hexose phosphate transporter UhpT (19, 20). The MICs of several antibiotics were also changed in the Δ*phoQ* mutant (Fig. 1). Among them, we focused on the increased susceptibility to minocycline and tigecycline in the Δ*phoQ* mutant (red bars). Although minocycline and tigecycline are potent antibiotics in treating infections caused by multidrug-resistant Gram-negative pathogens (21, 22), their mechanisms of resistance remain poorly understood, which prompted us to elucidate the underlying mechanisms of these phenotypes.

## PhoPQ two-component system is required for resistance to tetracycline and glycylcycline antibiotics

The MIC of minocycline was 4-fold lower in Δ*phoQ* mutant than in the wild-type strain, but the other mutant strains did not exhibit any changes (Fig. 2A). Minocycline and doxycycline belong to the tetracycline class, whereas tigecycline is a member of the glycylcycline class, which is derived from tetracycline (13). Therefore, the structural differences among these antibiotics (tetracycline, minocycline, doxycycline, and tigecycline) were not significant (Fig. 2B). The MICs of doxycycline, tigecycline, and tetracycline in the Δ*phoQ* mutant were 4-fold, 2-fold, and 1.5-fold lower, respectively, than in the wild-type strain (Fig. 2C; Fig. S3). We examined the effect of PhoP on antibiotic resistance as the sensor kinase PhoQ acts along with the response regulator PhoP. The MICs of the four antibiotics in the Δ*phoP* mutant were identical to those in the Δ*phoQ* mutant (Fig. 2C). The antibiotic sensitivity of the Δ*phoP* mutant was complemented by the pACYC184 plasmid-based expression of the *phoP* gene (Fig. 2D). These results

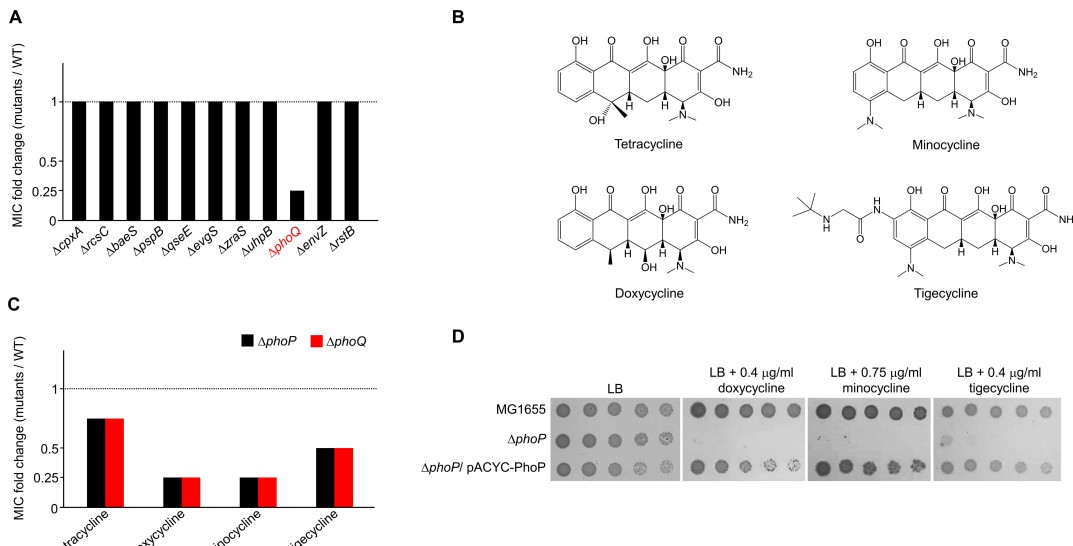

**FIG 2** The loss of the PhoPQ two-component system confers increased susceptibility to tetracycline and glycylcycline antibiotics. (A) Increased susceptibility of the ΔphoQ mutant to minocycline. The MICs of minocycline were measured against the wild-type and indicated mutant strains in MH medium. The relative MIC values for the indicated mutant cells compared to those for the wild-type cells are presented. (B) Structures of tetracycline and glycylcycline antibiotics. (C) Increased susceptibility of the ΔphoP or ΔphoQ mutant to tetracycline and glycylcycline antibiotics. The MICs of indicated antibiotics were measured against the wild-type and ΔphoP or ΔphoQ mutant strains in MH medium. The relative MIC values for the ΔphoP (black bars) or ΔphoQ (red bars) mutant cells compared to those for the wild-type cells are presented. (D) Complementation of antibiotic sensitivities of the ΔphoP mutant. The cells of the indicated strains were serially diluted from $10^8$ to $10^4$ cells/mL in 10-fold steps and spotted onto LB plates with or without the indicated concentration of each antibiotic. The experiments were performed in triplicate, and a representative image is presented.

indicate that the PhoPQ two-component system is necessary for intrinsic resistance to tetracycline and glycylcycline antibiotics.

## PhoPQ-mediated regulation of the expression of phosphoethanolamine transferase EptB affects intrinsic resistance to doxycycline and minocycline

PhoPQ two-component system regulates the transcriptional expression of diverse genes in response to magnesium starvation and acidic or antimicrobial peptide stress (3–5, 10). The magnesium transporter MgtA is a representative member of the PhoPQ regulon, and phosphorylated PhoP activates the transcription of the *mgtA* gene (7, 8). Therefore, we examined the effect of MgtA on doxycycline and minocycline resistance. The Δ*mgtA* mutant was not sensitive to doxycycline or minocycline (Fig. 3A), and the expression of *mgtA* in the Δ*phoP* mutant did not restore its sensitivity to minocycline (Fig. 3B). These results indicate that the minocycline sensitivity of the Δ*phoP* mutant was not associated with MgtA. Phenotypic results of another magnesium transporter, CorA, were almost identical to those of MgtA (Fig. 3A and B), implying that the minocycline sensitivity of the Δ*phoP* mutant was not caused by a defect in magnesium transportation. Subsequently, doxycycline resistance effects of several additional genes (*ompT*, *borD*, *pagP*, *tolC*, and *fadL*), whose transcription was activated by phosphorylated PhoP, were tested. The expression of these genes did not restore the sensitivity of the Δ*phoP* mutant to doxycycline (Fig. 3C).

Finally, to identify a gene associated with doxycycline sensitivity of the Δ*phoP* mutant, we used random transposon mutagenesis to screen for suppressors, in which the sensitivity of the Δ*phoP* mutant to doxycycline is restored. We isolated a suppressor, in which the sensitivity to doxycycline had recovered to almost the level of the wild-type strain (Fig. 4A). The transposon insertion was mapped within *eptB* encoding the phosphoethanolamine transferase that catalyzes the addition of phosphoethanolamine to the 3-deoxy-D-manno-oct-2-ulosonate (KdoII) in the inner core of LPS (23) (Fig. 4B; Fig. S4). We constructed a Δ*phoP* Δ*eptB* double mutant to confirm the effect of transposon

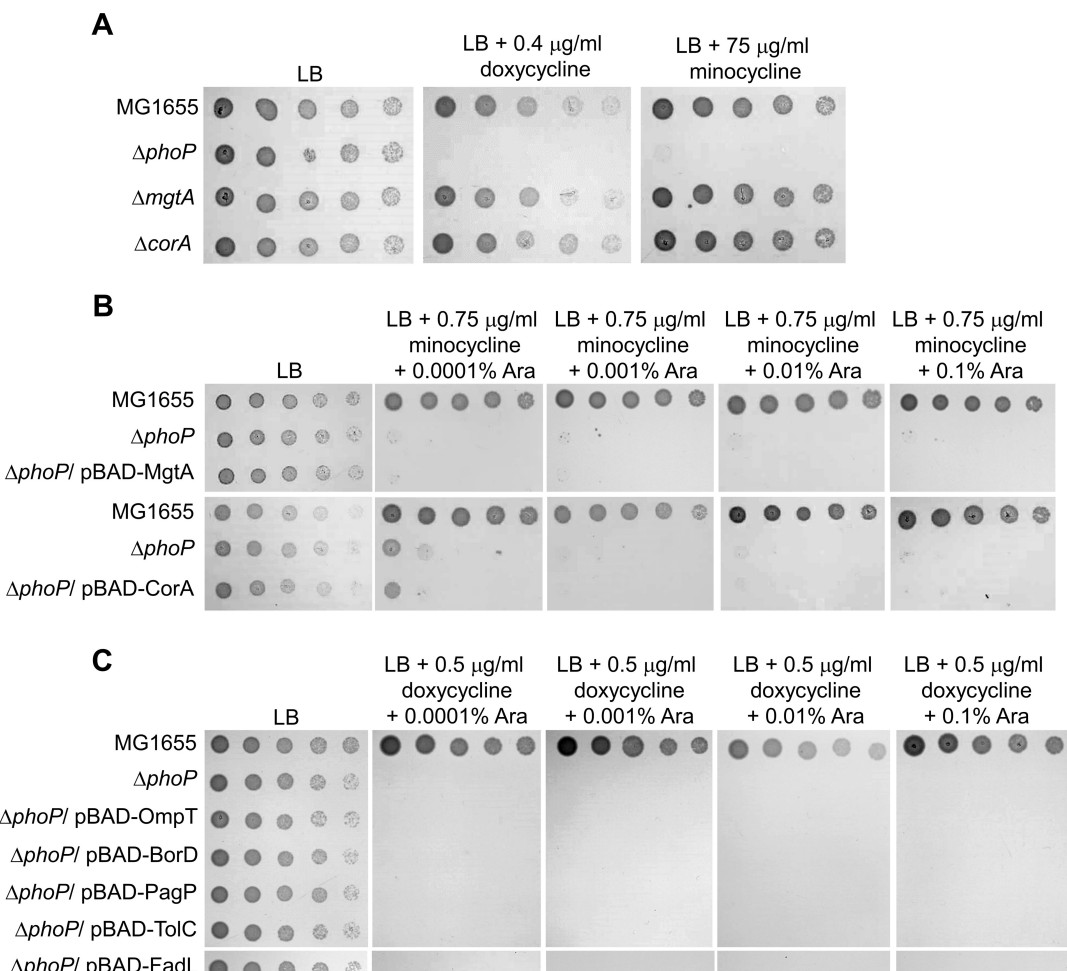

**FIG 3** The effect of several known PhoPQ regulon genes on the resistance to tetracycline antibiotics. (A) The effect of depletion of magnesium transporter genes on doxycycline and minocycline resistance. The cells of the indicated strains were serially diluted from $10^8$ to $10^4$ cells/mL in 10-fold steps and spotted onto LB plates with or without the indicated concentration of each antibiotic. (B) The effect of expression of magnesium transporter genes on minocycline resistance in the Δ*phoP* mutant strain. The cells of the indicated strains were serially diluted from $10^8$ to $10^4$ cells/mL in 10-fold steps and spotted onto LB plates with or without the indicated concentrations of minocycline and arabinose (Ara). (C) The effect of expression of several known PhoPQ regulon genes on doxycycline resistance in the Δ*phoP* mutant strain. The cells of the indicated strains were serially diluted from $10^8$ to $10^4$ cells/mL in 10-fold steps and spotted onto LB plates with or without the indicated concentrations of doxycycline and arabinose (Ara). (A–C) The experiments were performed in triplicate, and a representative image is presented.

insertion. The bacterial growth of this double mutant in the presence of doxycycline or minocycline was significantly restored compared to that of the Δ*phoP* mutant (Fig. 4C), thereby confirming that the deletion of the *eptB* gene suppresses the doxycycline sensitivity of the Δ*phoP* mutant. The expression of the *eptB* gene is silenced by the small regulatory RNA MgrR, whose transcription is activated by phosphorylated PhoP (24, 25). Therefore, the expression of the *eptB* gene could be activated in the Δ*phoP* mutant, which may cause doxycycline sensitivity. To assess this assumption, we measured the transcription of *eptB* and *phoP* in the wild-type and Δ*phoP* mutant strains. Expectedly, the *phoP* transcripts were not detected in the Δ*phoP* mutant, and the level of the *eptB* transcripts was almost 4-fold higher in the Δ*phoP* mutant than in the wild-type strain (Fig. 4D). Additionally, the overexpression of *eptB* in the wild-type strain using the plasmid pBAD24 induced significant sensitivity to doxycycline (Fig. 4E). Overall, these results demonstrated that the PhoPQ-mediated regulation of *eptB* expression is associated with doxycycline and minocycline resistance.

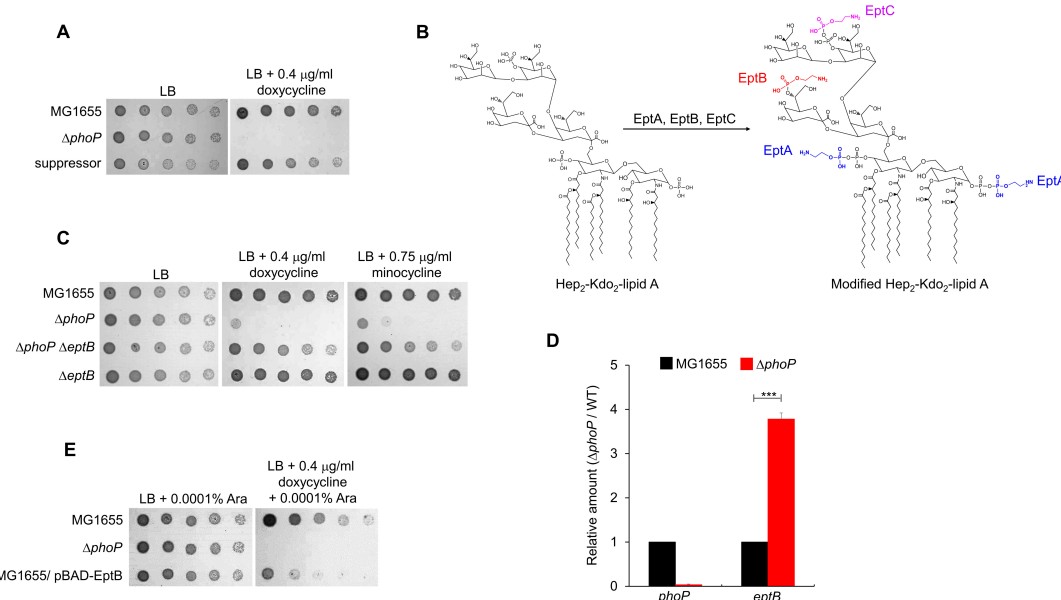

**FIG 4** The effect of the phosphoethanolamine transferase EptB on the resistance to tetracycline antibiotics. (A) Isolation of the suppressor mutant of the $\Delta phoP$ mutant. The cells of the indicated strains were serially diluted from $10^8$ to $10^4$ cells/mL in 10-fold steps and spotted onto LB plates with or without the indicated concentration of doxycycline. (B) Schematic representation depicting the roles of phosphoethanolamine transferases in LPS modification. EptA, EptB, and EptC catalyze the addition of phosphoethanolamine to the phosphate group of the glucosamine disaccharide of lipid A, the KdoII sugar in the inner core, and the phosphate group of the heptose I residue in the inner core, respectively. (C) The depletion of EptB suppresses the sensitivity of the $\Delta phoP$ mutant to tetracycline antibiotics. The cells of the indicated strains were serially diluted from $10^8$ to $10^4$ cells/mL in 10-fold steps and spotted onto LB plates with or without the indicated concentration of doxycycline or minocycline. (D) Relative mRNA levels of the $phoP$ and $eptB$ genes in the wild-type and $\Delta phoP$ mutant strains. Total mRNA was extracted from the wild-type (black bars) and $\Delta phoP$ mutant (red bars) cells cultured up to the early exponential phase [the optical density at 600 nm $(OD_{600}) = 0.4$]. mRNA levels of the $phoP$ and $eptB$ genes were normalized to the levels of 16S rRNA. Data were produced from three independent experiments. Statistical significance was determined using the student's $t$-test. ***$P < 0.001$. (E) The effect of overexpression of EptB on doxycycline resistance. The cells of the indicated strains were serially diluted from $10^8$ to $10^4$ cells/mL in 10-fold steps and spotted onto LB plates with or without the indicated concentrations of doxycycline and arabinose (Ara). (A, C, and E) The experiments were performed in triplicate, and a representative image is presented.

## Other phenotypes of the *phoP* mutant were not restored by the deletion of *eptB*

To assess the effect of EptB on other phenotypes of the PhoPQ two-component system, we examined additional phenotypes of the $\Delta phoP$ or $\Delta phoQ$ mutant. First, the effect of the PhoPQ two-component system on bacterial growth under various stress conditions was examined. The $\Delta phoP$ and $\Delta phoQ$ mutants were significantly sensitive to various stresses, including sodium dodecyl sulfate/EDTA, bile salt, EDTA, and acidic pH (Fig. 5A). We assessed whether the deletion of the $eptB$ gene could suppress these phenotypes of the $\Delta phoP$ mutant, as in the case of minocycline and doxycycline. Most phenotypes of the $\Delta phoP$ mutant were not restored by the deletion of the $eptB$ gene, although bile salt sensitivity of the $\Delta phoP$ mutant was slightly recovered (Fig. 5B). Next, we examined the MICs of various antibiotics against the $\Delta phoP$ mutant. The MICs of antibiotics for the $\Delta phoP$ mutant were similar to those for the $\Delta phoQ$ mutant (Fig. 1 and 5C). We examined the effect of EptB on antibiotic resistance in the $\Delta phoP$ mutant. Unlike minocycline and doxycycline, most phenotypes of the $\Delta phoP$ mutant against antibiotics were not restored by the deletion of the $eptB$ gene, although trimethoprim sensitivity of the $\Delta phoP$ mutant was slightly recovered (Fig. 5D). Collectively, these results showed that most phenotypes of the $\Delta phoP$ mutant were not restored by the deletion of the $eptB$ gene, indicating a specific relationship between EptB and the resistance to minocycline and doxycycline.

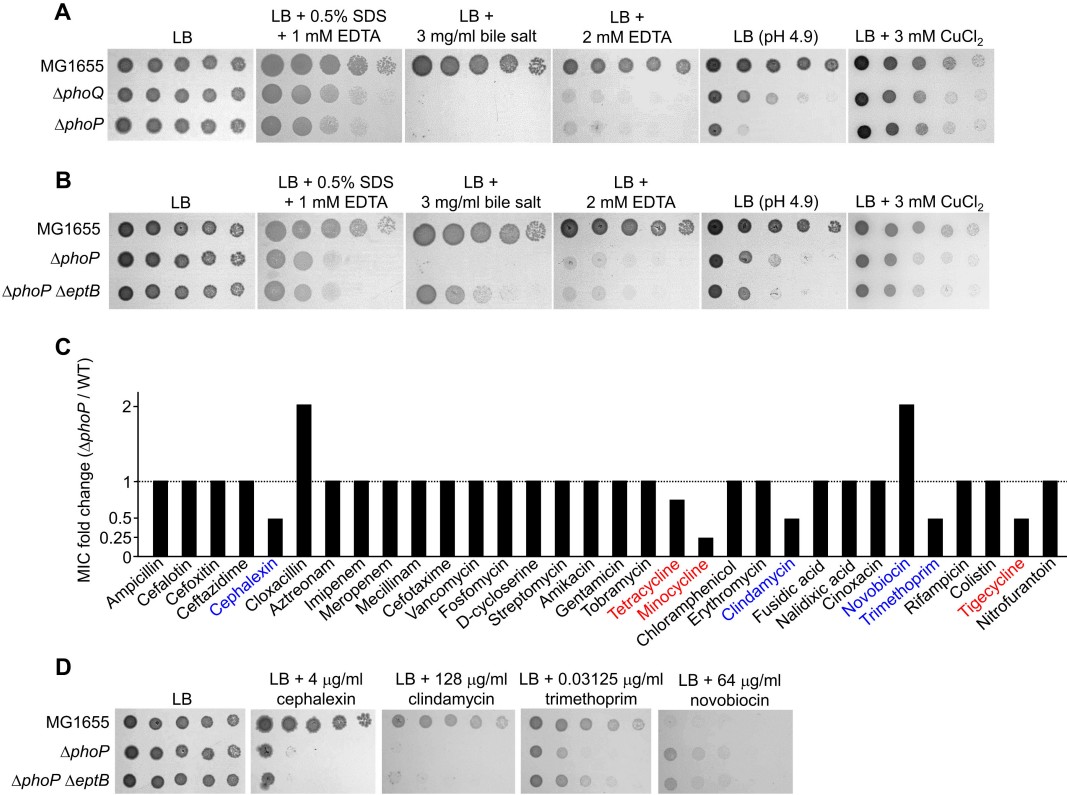

**FIG 5** The effect of EptB on various phenotypes of the Δ*phoP* mutant. (A) Growth defect of the Δ*phoP* and Δ*phoQ* mutant strains under various envelope stress conditions. The cells of the indicated strains were serially diluted from $10^8$ to $10^4$ cells/mL in 10-fold steps and spotted onto LB plates with or without the indicated chemicals, or an acidic LB plate. (B) The effect of EptB inactivation on the sensitivity of the Δ*phoP* mutant to envelope stress. The cells of the indicated strains were serially diluted from $10^8$ to $10^4$ cells/mL in 10-fold steps and spotted onto LB plates with or without the indicated chemicals, or an acidic LB plate. (C) The effect of PhoP on the MICs of antibiotics. The MICs of various antibiotics were measured against the wild-type and Δ*phoP* mutant strains in MH medium. The relative MIC values for the Δ*phoP* mutant cells compared to those for the wild-type cells are presented. (D) The effect of EptB inactivation on the altered susceptibility of the Δ*phoP* mutant against antibiotics. The cells of the indicated strains were serially diluted from $10^8$ to $10^4$ cells/mL in 10-fold steps and spotted onto LB plates with or without the indicated concentrations of antibiotics. (A, B, and D) The experiments were performed in triplicate, and a representative image is presented.

## Depletion of EtpB alleviates increased permeability of the *phoP* mutant to doxycycline

LPS is present in the outer membrane, and its modification by EptB can affect the penetration of antibiotics through the outer membrane. Therefore, we measured the penetration levels of doxycycline in the wild-type and mutant strains. Intracellular doxycycline accumulation was measured using ELISA. The Δ*phoP* mutant showed increased intracellular levels of doxycycline compared to the wild-type strain (Fig. 6). Increased levels of doxycycline in the Δ*phoP* mutant were complemented by pACYC184 plasmid-based expression of the *phoP* gene. Deletion of the *eptB* gene also diminished the increased intracellular levels of doxycycline in the Δ*phoP* mutant (Fig. 6). Therefore, these results implied that the doxycycline sensitivity of the Δ*phoP* mutant was at least partially due to the increased permeability of doxycycline.

## EptB is associated with resistance to tetracycline and glycylcycline antibiotics

Two other phosphoethanolamine transferases are involved in LPS modification in *E. coli*, in addition to EptB. EptA catalyzes the addition of phosphoethanolamine to the phosphate group of glucosamine disaccharide of lipid A (26, 27), whereas EptC catalyzes the incorporation of phosphoethanolamine into the phosphate group of the heptose I

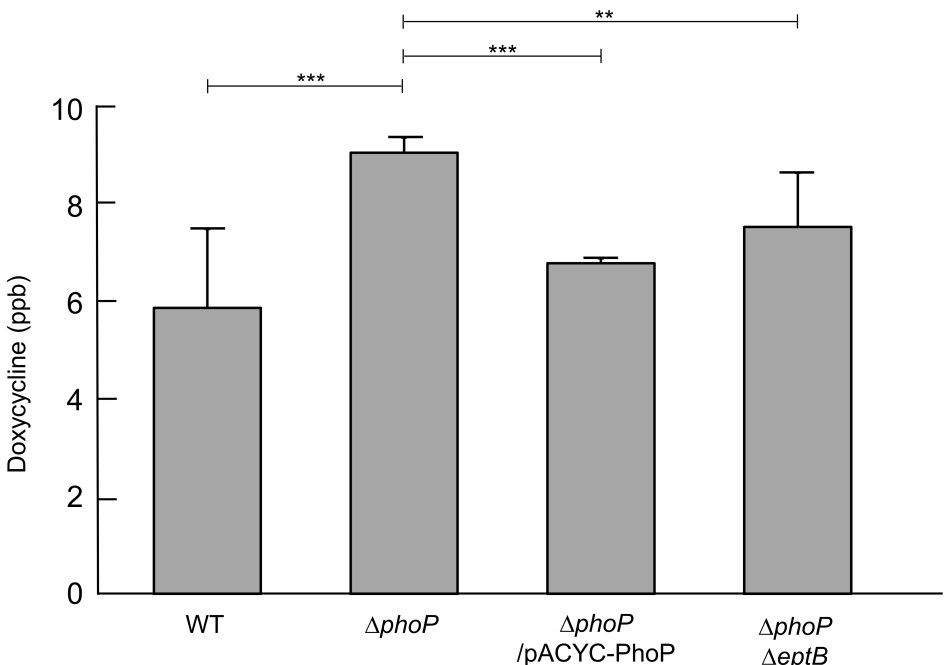

**FIG 6** The intracellular accumulated levels of doxycycline. At the early exponential phase, 0.5 mg/mL of doxycycline was added to LB medium and the cells were harvested after additional incubation for 20 min at 37°C. After washing, the harvested cells were disrupted and cell debris was removed by centrifugation. After removing soluble proteins using acetonitrile, the doxycycline level in the supernatant was determined using a Doxycycline ELISA Kit. The doxycycline level was estimated by measuring the absorbance at 450 nm. The exact concentration of doxycycline was estimated based on the standard curve made using the standard concentrations of doxycycline. Data were produced from three independent experiments. Statistical significance was determined using the student's $t$-test. **$P < 0.01$; ***$P < 0.001$.

residue in the inner core (28) (Fig. 4B). We examined whether the deletion of *eptA* or *eptC* suppresses doxycycline sensitivity of the Δ*phoP* mutant. Unike EptB, the depletion of EptA or EptC did not suppress doxycycline sensitivity of the Δ*phoP* mutant (Fig. 7A), indicating a distinct role of EptB in doxycycline resistance. We constructed single-deletion mutants of each gene and examined the MICs of various antibiotics in each mutant strain to analyze the roles of the three phosphoethanolamine transferases in more detail. Notably, neither Δ*eptA* nor Δ*eptC* mutants revealed any change in the MICs of the antibiotics tested (Fig. S5). Meanwhile, the Δ*eptB* mutant showed 2-fold and 1.5-fold higher MICs of minocycline and tigecycline, respectively, than the wild-type strain, whereas there were no changes in the MICs of other antibiotics tested (Fig. 7B). When the bacterial growth of the three mutants was measured in the presence of antibiotics, the Δ*eptB* mutant exhibited resistance to minocycline, tigecycline, and doxycycline, whereas the growth levels of the Δ*eptA* and Δ*eptC* mutants were identical to those of the wild-type strain for all the antibiotics tested (Fig. 7C). In conclusion, our results demonstrate that EptB depletion induces resistance to tetracycline and glycylcy-cline antibiotics.

## DISCUSSION

Antibiotic resistance of Gram-negative pathogens poses a serious threat on public health worldwide (12, 29–31). Tetracycline and glycylcycline antibiotics, especially tigecycline, are among the important therapeutic options for treating infections caused by multidrug-resistant Gram-negative pathogens (12, 30). Despite the clinical importance of tetracycline and glycylcycline antibiotics, the molecular mechanisms underlying the resistance to these antibiotics are not fully understood. In this study, we demonstrated that PhoPQ-mediated modification of the core region of LPS affects

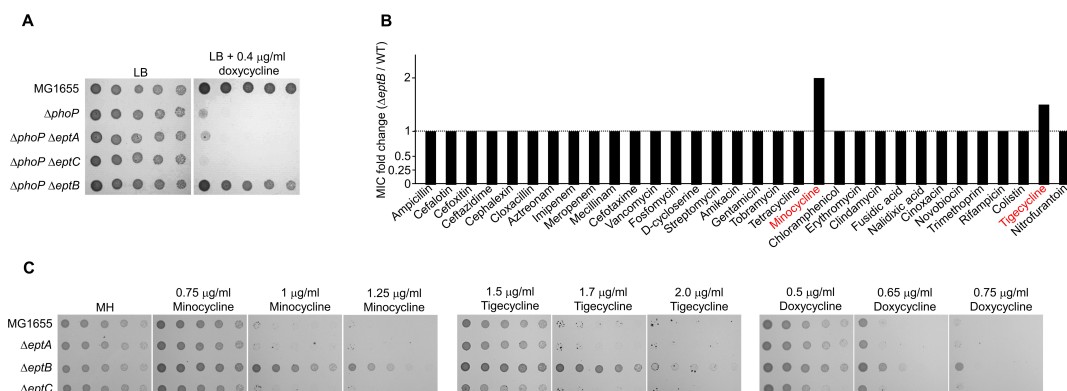

**FIG 7** The depletion of EptB induces tetracycline and glycylcycline resistance. (A) The effect of inactivation of the phosphoethanolamine transferases on the sensitivity of the Δ*phoP* mutant to doxycycline. The cells of the indicated strains were serially diluted from $10^8$ to $10^4$ cells/mL in 10-fold steps and spotted onto LB plates with or without doxycycline. (B) The effect of EptB depletion on the MICs of antibiotics. The MICs of various antibiotics were measured against the wild-type and Δ*eptB* mutant strains in MH medium. The relative MIC values for the Δ*eptB* mutant cells compared to those for the wild-type cells are presented. (C) The resistance of the Δ*eptB* mutant to tetracycline and glycylcycline antibiotics. The cells of the indicated strains were serially diluted from $10^8$ to $10^4$ cells/mL in 10-fold steps and spotted onto LB plates with or without the indicated concentration of each antibiotic. The experiments were performed in triplicate, and a representative image is presented.

resistance to tetracycline and glycylcycline antibiotics. Phosphoethanolamine transferase EptB, a member of the PhoPQ regulon, adds phosphoethanolamine to the KdoII sugar in the inner core of LPS, which induces sensitivity to tetracycline and glycylcycline antibiotics (Fig. 2 to 4). These phenotypes appear to be caused by the increased uptake of tetracycline and glycylcycline antibiotics (Fig. 6). Modification of the inner core of LPS by EptB affected susceptibility to tetracycline and glycylcycline antibiotics, among many antibiotics with diverse modes of action (Fig. 7). These results revealed the novel physiological significance of PhoPQ-mediated modification of the inner core of LPS.

Several regions of LPS can be modified by adding a phosphoethanolamine group (23, 26–28). Among them, the addition of phosphoethanolamine to the phosphate group of the glucosamine disaccharide of lipid A has been extensively studied (32–34). Attachment of the phosphoethanolamine group neutralizes the negative charge of the lipid A phosphate group, which inhibits binding of LPS to colistin and cationic antimicrobial peptides (33–35). Consequently, this modification results in increased resistance to colistin and cationic antimicrobial peptides. The addition of other functional groups such as aminoarabinose (36), glucosamine (37), galactosamine (38, 39), and glucose (38), which can neutralize the negative charge of the lipid A phosphate group, also induces similar resistance. In contrast to lipid A modifications, studies on core modifications are scarce. Phosphoethanolamine addition to the phosphate group of heptose I in the inner core by EptC is required to overcome envelope stresses such as SDS and $Zn^{2+}$ (28). In *E. coli*, the deletion of EptC resulted in slightly increased susceptibility to polymyxin B (40). However, the physiological role of the EptB-dependent phosphoethanolamine addition to KdoII in the inner core has not yet been elucidated. Our study demonstrated that EptB overexpression increased susceptibility to tetracycline and glycylcycline antibiotics, whereas the deletion of the *eptB* gene resulted in elevated resistance to these antibiotics. These results indicate that EptB-dependent phosphoethanolamine addition to KdoII is associated with resistance to tetracycline and glycylcycline antibiotics. Notably, neither EptA nor EptC affected resistance to tetracycline and glycylcycline antibiotics, and EptB did not affect colistin resistance (Fig. 7), indicating the distinct roles among EptA, EptB, and EptC. To the best of our knowledge, this is the first report demonstrating the physiological role of KdoII modification in the inner core of LPS.

Most studies on LPS modification have been focused on the neutralization of phosphate groups present in lipid A or the core region. A reduction in the negative charge of the phosphate group of lipid A decreases the electrostatic interactions

between colistin and lipid A, which induces colistin resistance (41). Meanwhile, the addition of the phosphoethanolamine group to the hydroxyl group of KdoII by EptB did not induce a reduction in the formal charge of LPS (Fig. 4B). Therefore, the deletion of EptB did not affect susceptibility to colistin (Fig. 7B). The MICs of all the antibiotics tested, except minocycline and tigecycline, were not changed by EptB deletion (Fig. 7B). These results indicate that EptB-mediated KdoII modification is involved in resistance to tetracycline and glycylcycline antibiotics. The mechanisms by which tetracycline and glycylcycline antibiotics penetrate the outer membrane are poorly understood. As EptB-mediated KdoII modification affects the resistance to various tetracycline and glycylcycline antibiotics with different side-chain functional groups (Fig. 1B and 7), the penetration of a linearly fused tetracyclic nucleus across the outer membrane may be affected by KdoII modification. Notably, the deletion of *eptB* significantly suppressed the sensitivity of the *ΔphoP* mutant to bile salts (Fig. 5B). Like tetracycline and glycylcycline antibiotics, bile salts also possess a similar fused tetracyclic nucleus (42). Therefore, the relationship between the penetration of molecules with a fused tetracyclic nucleus across the outer membrane and KdoII modification can be investigated in further studies.

Various resistance mechanisms against tetracycline and glycylcycline antibiotics have been unveiled, such as tetracycline-specific Tet(A) or Tet(B) efflux pump acquisition (11), GTP-dependent release mechanisms of tetracycline from the ribosome by Tet(M) or Tet(O) (11, 43, 44), and enzymatic inactivation of tetracycline by Tet(X) or Tet(37) (45, 46). Loss of the OmpF and OmpC porins induces intrinsic tetracycline resistance via reduced transportation of tetracycline across the outer membrane (47, 48). In this study, we revealed a novel intrinsic resistance mechanism for tetracycline and glycylcy-cline antibiotics. The phosphoethanolamine addition to the core sugar of LPS enhanced susceptibility to tetracycline and glycylcycline antibiotics (Fig. 4), whereas the loss of this modification induced intrinsic resistance to these antibiotics (Fig. 7). These effects may be caused by the altered transportation of these antibiotics across the outer membrane (Fig. 6). Therefore, our study demonstrated that LPS core modification is a novel intrinsic resistance mechanism against tetracycline and glycylcycline antibiotics.

The PhoPQ system is a two-component system that has been extensively studied in various Gram-negative bacteria (49, 50). Diverse signals that activate the sensor kinase PhoQ have been identified, including $Mg^{2+}$ and other divalent cations (3), antimicrobial peptides (5), mildly acidic pH (4, 10), osmotic upshift (6), and long-chain unsaturated fatty acids (51). Activated response regulator PhoP regulates the transcription of many genes and diverse proteins via the regulation of other transcriptional factors, protease regulators, metabolites, and regulatory RNAs (50). These PhoP-mediated changes cause diverse phenotypic consequences in many bacterial behaviors, such as metal ion homeostasis, virulence, motility, resistance to antimicrobial agents, and the ability to overcome stressful conditions such as acidic stress or nutritional depletion (49, 50). In *E. coli*, PhoP regulates the transcription of several genes associated with LPS modification, such as *eptA* and *eptB* encoding the phosphoethanolamine transferase (24, 52) and *pagP* encoding the lipid A palmitoyl transferase (53). EptA induces colistin resistance via neutralization of the phosphate groups present in lipid A (33–35), whereas PagP induces resistance to cationic alpha-helical antimicrobial peptides such as C18G, magainin 2, and cecropin A (50, 54, 55). However, the physiological significance of EptB remains unclear. Here, we revealed that EptB affects resistance to tetracycline and glycylcycline antibiotics. These results demonstrate that all PhoPQ-mediated LPS modifications are associated with antibiotic resistance.

Systematic analysis of the two-component systems associated with the envelope stress response showed that several two-component systems affect intrinsic resistance to various antibiotics. The *ΔenvZ* mutant was sensitive to vancomycin, fosfomycin, D-cycloserine, streptomycin, amikacin, tobramycin, and novobiocin, besides β-lactams (Fig. 1). Because OmpC and OmpF porins regulated by the EnvZ-OmpR two-component system act as a channel for the influx of various antibiotics and OmpC is also impor-tant for the maintenance of membrane integrity (18), these phenotypes of the *ΔenvZ*

mutant appear to be caused by altered expression levels of OmpC and OmpF. The ΔcpxA mutant was sensitive to chloramphenicol, erythromycin, nalidixic acid, tigecycline, and nitrofurantoin (Fig. 1). In *Klebsiella pneumoniae*, the sensitivity of the ΔcpxA mutant to chloramphenicol was reported (56), but its exact mechanism was not determined. Similarly, in *Haemophilus parasuis*, the sensitivity of the ΔcpxA mutant to erythromycin was reported (57), but its exact mechanism was not revealed. Notably, a recent study showed that CpxAR-mediated expressions of nitroreductases affect the prodrug activation of nitrofurantoin (58). Therefore, the sensitivity of the ΔcpxA mutant to nitrofurantoin could be caused by altered expression of nitroreductases. The ΔphoQ mutant was sensitive to cephalexin, erythromycin, and trimethoprim, besides tetracycline and tigecycline, whereas it was resistant to cloxacillin, cinoxacin, and novobiocin (Fig. 1). A recent study demonstrated that the PhoPQ system and MgrB affect susceptibility to trimethoprim by modulating the expression of dihydrofolate reductase, a target protein of trimethoprim (59). The relationship between the PhoPQ system and other antibiotics identified in this study has not been reported yet. Therefore, further studies are necessary to elucidate the molecular mechanisms underlying the resistance of the ΔphoQ mutant to these antibiotics.

## MATERIALS AND METHODS

### Bacterial strains, plasmids, and culture conditions

All bacterial strains and plasmids used in this study are presented in Table S1, and all primers are listed in Table S2. Bacterial cells were cultured in Luria–Bertani (LB) medium at 37°C, unless otherwise mentioned. Antibiotics, including kanamycin (50 mg/mL), chloramphenicol (5 mg/mL), tetracycline (10 mg/mL), and ampicillin (100 mg/mL), were added to the culture medium as required. Bacterial growth was estimated using a serial dilution spotting assay onto LB agar plates. Cells from overnight cultures in LB medium were inoculated into fresh LB medium. On reaching an $OD_{600}$ of approximately 0.8, the cultured cells were serially diluted 10-fold from $10^8$ to $10^4$ cells/mL, and 2 mL of the samples was spotted onto LB agar plates with or without the indicated chemicals. After incubation at 37°C for 10–20 h, plates were imaged using a digital camera, EOS 100D (Canon Inc., Japan).

All deletion mutants were constructed using λ red recombinase, as described previously (60). DNA products for gene deletion were prepared by polymerase chain reaction (PCR) using primers with 50 bp sequence for homologous recombination, and the plasmid pKD13 with a kanamycin resistance gene as a template. After the purification of the PCR products, the purified deletion cassettes were electroporated into MG1655 or mutant cells harboring the plasmid pKD46 expressing λ red recombinase. Deletion mutants were selected on LB plates containing kanamycin, and the deletion of the target gene was confirmed using PCR. The kanamycin resistancegene inserted into the chromosome was removed by the plasmid pCP20 expressing the FLP recombinase, as described previously (60). The plasmid pCP20 in the mutant cells was removed by incubation at 37°C, instead of incubation at 42°C, to decrease physiological changes in the bacteria cells (61, 62).

DNA covering both the promoter region and open reading frame of PhoP (from −170 to +690) was cloned into the plasmid pACYC184. PCR was performed by using a forward primer possessing a synthetic BamHI site (underlined) (5′- CCCGTCCTGT<u>GGATCC</u>AAACCT CGTATCAGTGCCGG-3′), and a reverse primer possessing a synthetic EagI site (underlined) (5′- CCCAGCGCGT<u>CGGCCG</u>GACGCAGTAATTTTTTCATC-3′). PCR product was inserted into the plasmid pACYC184 digested using BamHI and EagI by infusion cloning (Clontech, USA), as reported previously (63). Cloning was confirmed by PCR using other primer sets located within the plasmid pACYC184 and DNA sequencing. pBAD24 plasmid-expressing genes regulated by the PhoPQ system, such as *mgtA*, were constructed using a similar method. The entire open reading frame of each gene was amplified using PCR, and the PCR product was inserted into the plasmid pBAD24 digested by EcoRI and XbaI, through

homologous recombination between overlapping sequences using infusion cloning. Target gene cloning was confirmed using DNA sequencing.

## Determination of MICs of antibiotics

The MICs of antibiotics were determined according to the guidelines provided by the Clinical and Laboratory Standards Institute (64). All wild-type and mutant strains were cultured overnight in Mueller–Hinton (MH) broth and subsequently inoculated into fresh MH medium. When the bacterial suspensions reached a turbidity of 0.5 McFarland standard (approximately $1.5 \times 10^8$ cells/mL), the cells were diluted to a final concentration of $10^7$ cells/mL using MH broth. A diluted suspension of 10 mL was spotted onto MH plates containing antibiotics at final concentrations ranging from 1,024 to 7.8 ng/mL in 2-fold serial dilutions. The MIC of each antibiotic was determined after incubation at 37°C for 20 h, based on the bacterial growth. The MIC corresponds to the lowest concentration at which visible lawn growth of the cell spot is inhibited.

## Transposon mutagenesis and identification of transposon insertion site

Transposon mutagenesis was performed to identify a mutant that suppresses the doxycycline sensitivity of the Δ*phoP* mutant, using the *pir*-dependent transposon delivery vector pRL27 carrying a Tn5 transposase gene and a mini-Tn5 element encoding kanamycin resistance (65). The pRL27 plasmid was amplified in *E. coli* DH5αλ*pir* cells carrying the *pir* gene. Purified pRL27 plasmids were electroporated into competent cells of the Δ*phoP* mutant. The pRL27 plasmid was not replicated in this strain as the Δ*phoP* mutant did not harbor the *pir* gene; therefore, the kanamycin resistance gene in this strain was maintained when the chromosomal insertion of mini-Tn5 element occurred. The mutant that suppressed doxycycline sensitivity of the Δ*phoP* mutant was selected using an LB plate containing both kanamycin (50 mg/mL) and doxycycline (0.4 mg/mL). PCR was performed to uncover the transposon insertion site, using the genomic DNA of the suppressor strain as a template and primer sequences (an arbitrary primer consisting of a GGCGGT sequence and a random sequence, and a Tn5 transposon inner primer, 5′-GGTTGTAACACTGGCAGAGCATTACG-3′), as described previously (66). After PCR purification, product was sequenced using another Tn5 transposon inner primer, 5′-ATCA GCAACTTAAATAGCCTCTAAGG-3′.

## Quantitative real-time RT-PCR

Wild-type and Δ*phoP* mutant strains cultured overnight in LB medium were inoculated into fresh LB medium and cultured at 37°C to reach early exponential phase. Total RNA was extracted from the cells using the RNeasy Mini Kit (Qiagen, USA). Contaminating DNA in the samples were removed through incubation at 37°C for 2 h using RNase-free DNase I (Promega, USA). All RNAs in the samples were converted into cDNA using a cDNA EcoDry Premix (Clontech, USA). cDNA levels of the *phoP* and *eptB* genes were quantified by PCR using SYBR Premix Ex Taq II (Takara, Japan) solution containing RT-PCR primers for each gene (See Table S2) and 10-fold diluted cDNA samples as templates, in a CFX96 Real-Time System (Bio-Rad, USA). The 16S rRNA gene was used as the reference to estimate the expression level of each gene.

## Estimation of doxycycline uptake

Cells cultured overnight in LB medium were inoculated into fresh LB medium. At the early exponential phase, 0.5 mg/mL of doxycycline was added into the LB medium and cells were cultured for 20 min at 37°C. Cells were harvested and washed using washing buffer [100 mM Tris-HCl (pH 8.0), and 100 mM NaCl]. The cells were resuspended in 1 mL of resuspension buffer [50 mM Tris-HCl (pH 8.0), and 300 mM NaCl] and disrupted using a French press at 8,000 psi. The sample was centrifuged at $10,000 \times g$ for 10 min at 4°C, and the supernatant was mixed with 1 mL of acetonitrile. Precipitated proteins were removed by centrifugation at $10,000 \times g$ for 5 min at 4°C and the supernatant was diluted

5-fold using distilled water. Doxycycline levels in diluted samples were determined using a Doxycycline ELISA Kit (BioVision, USA). Doxycycline levels were estimated by measuring the absorbance at 450 nm. The exact concentration of doxycycline was estimated based on a standard curve of doxycycline drawn using standard concentrations.

## ACKNOWLEDGMENTS

This work was supported by research grants from Basic Science Research Program through the National Research Foundation of Korea funded by the Ministry of Education (NRF-RS-2023-00246684) and Korea Institute of Planning and Evaluation for Technology in Food, Agriculture and Forestry through High Value-added Food Technology Development Program, funded by Ministry of Agriculture, Food and Rural Affairs (grant number 322026-3).

## AUTHOR AFFILIATION

[1]Department of Biological Sciences, Myongji University, Yongin, Gyeonggido, Republic of Korea

## AUTHOR ORCIDs

Chang-Ro Lee  http://orcid.org/0000-0002-9088-4262

## FUNDING

| Funder | Grant(s) | Author(s) |
| --- | --- | --- |
| Basic Science Research Program | RS-2023-00246684 | Chang-Ro Lee |
| High Value-added Food Technology Development Program | 322026-3 | Chang-Ro Lee |

## AUTHOR CONTRIBUTIONS

Byoung Jun Choi, Data curation, Formal analysis, Investigation, Methodology, Software, Validation, Visualization, Writing – review and editing | Umji Choi, Data curation, Formal analysis, Funding acquisition, Investigation, Methodology, Resources, Software, Supervision, Validation, Writing – review and editing | Dae-Beom Ryu, Investigation, Methodology, Resources, Software, Writing – review and editing | Chang-Ro Lee, Conceptualization, Data curation, Formal analysis, Funding acquisition, Investigation, Methodology, Project administration, Resources, Supervision, Validation, Writing – original draft, Writing – review and editing

## ADDITIONAL FILES

The following material is available online.

### Supplemental Material

**Supplemental material (mSystems00964-24-s0001.docx).** Supplemental figures and tables.

### Open Peer Review

**PEER REVIEW HISTORY (review-history.pdf).** An accounting of the reviewer comments and feedback.

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
