## [Reviewer comments · mSystems]

PhoPQ-mediated lipopolysaccharide modification governs intrinsic resistance to tetracycline and glycylicline antibiotics in *Escherichia coli*

Chang-Ro Lee, Byoung Jun Choi, Umji Choi, and Dae-Beom Ryu

Corresponding Author(s): Chang-Ro Lee, Myongji University - Natural Science Campus

Review Timeline:

Submission Date:	July 17, 2024
Editorial Decision:	August 25, 2024
Revision Received:	September 1, 2024
Accepted:	September 8, 2024

Editor: Mehrad Hamidian

Reviewer(s): Disclosure of reviewer identity is with reference to reviewer comments included in decision letter(s). The following individuals involved in review of your submission have agreed to reveal their identity: You-Hee Cho (Reviewer #1)

Transaction Report:

DOI: <https://doi.org/10.1128/msystems.00964-24>

Re: mSystems00964-24 (PhoPQ-mediated lipopolysaccharide modification regulates intrinsic resistance to tetracycline and glycylicyline antibiotics in *Escherichia coli*)

Dear Prof. Chang-Ro Lee:

- Introduction needs to be more concise.
- Tetracyclines can't be considered as last resort antibiotics.
- do mutants defective in sensor kinases grow normally? Did you test for this?
- Line 106: "various antibiotics", please specify the antibiotics.
- include the transposon mutagenesis library stats including the number of reads produces, coverage, saturation plots etc. Please include the stats in the text but the saturation plot should go to the supplementary material. As raised by the reviewer, this section also needs some work and clarification. In addition, sequencing reads should be submitted to GenBank and made publically available with the GenBank SRA numbers included in the revised manuscript.

Revision Guidelines

Sincerely,
Mehrad Hamidian
Editor
mSystems

Reviewer #1 (Comments for the Author):

In this paper by Choi et al., the authors demonstrated that PhoPQ-mediated modification of the LPS inner core contributes to intrinsic resistance to tetracycline antibiotics.

The paper is well organized and thus reads well with the emphasis of the role of EptB in the antibiotic resistance, which was identified as a phoP mutant suppressor in this study. The comprehensive genetic analyses in this study would be a major strength of this paper, which could connect the dots between the intrinsic antibiotic resistance and the PhoPQ TCS, based on the well-known aspects of regulation by the PhoPQ TCS. However, this brings up what I see as a limitation of the paper, since a couple of major concerns have been at the end, in that the intrinsic resistance to tetracycline antibiotics directed by upregulation of EptB could be observed under the physiological conditions of *E. coli*.

Nevertheless, the results are well presented with appropriate controls. It should be also noted that this study covers important aspects in microbiology, which include TCSs and LPS modification in regard to intrinsic antibiotic resistance. I would have some concerns as below, which need to be clarified to improve the current manuscript.

1. I would suggest the authors to describe "contributes" or "governs" etc rather than "regulates", given that they have no direct evidence of that intrinsic resistance is indeed regulated in a kind of adaptive response in *E. coli*.
2. Fig 4: Please designate the precise position of the Tn5 insertion within the ORF of eptB. I am just wondering whether or not the overexpression of EptB affects the resistance to other tetracyclines and tigecycline.
3. Discussion: In Fig. 1, three mutants (*envZ*, *phoQ*, and *cpxA* mutants) exhibited sensitivity to various antibiotics. The detailed explanation of the relevant mechanisms should be covered in the discussion section.
4. L282-285: Fig. 3B showed that the deletion of eptB significantly suppressed the sensitivity of the phoP mutant to bile salt. Like tetracycline and glycylicycline antibiotics, the bile salt also possesses a fused tetracyclic nucleus, which can be discussed in detail.

September 2, 2024

Dear Editor,

We are submitting the revised version of manuscript for publication in the “mSystems”. We greatly thank you and reviewers for their careful and thorough comments on our manuscript. We did our best to appropriately respond to their comments. We hope that our manuscript is suitable to be published in the journal.

We provide an itemized list of additional changes.

We appreciate the comments that reviewers have made regarding our manuscript (Manuscript ID: mSystems00964-24). We have carefully read the comments and the manuscript has been rewritten in response to these comments as detailed below.

I. Response to comment of Reviewer 1

In this paper by Choi et al., the authors demonstrated that PhoPQ-mediated modification of the LPS inner core contributes to intrinsic resistance to tetracycline antibiotics.

The paper is well organized and thus reads well with the emphasis of the role of EptB in the antibiotic resistance, which was identified as a phoP mutant suppressor in this study. The comprehensive genetic analyses in this study would be a major strength of this paper, which could connect the dots between the intrinsic antibiotic resistance and the PhoPQ TCS, based on the well-known aspects of regulation by the PhoPQ TCS. However, this brings up what I see as a limitation of the paper, since a couple of major concerns have been at the end, in that the intrinsic resistance to tetracycline antibiotics directed by upregulation of EptB could be observed under the physiological conditions of *E. coli*.

Nevertheless, the results are well presented with appropriate controls. It should be also noted that this study covers important aspects in microbiology, which include TCSs and LPS modification in regard to intrinsic antibiotic resistance. I would have some concerns as below, which need to be clarified to improve the current manuscript.

Major points.

1. I would suggest the authors to describe "contributes" or "governs" etc rather than "regulates", given that they have no direct evidence of that intrinsic resistance is indeed regulated in a kind of adaptive response in *E. coli*.

Response: In the entire manuscript, we changed “regulates” to “governs” or “affects”.

2. Fig 4: Please designate the precise position of the Tn5 insertion within the ORF of eptB. I am just wondering whether or not the overexpression of EptB affects the resistance to other tetracyclines and tigecycline.

Response: We added a novel figure (Fig. S4) showing the precise position of the Tn5 insertion.

3. Discussion: In Fig. 1, three mutants (*envZ*, *phoQ*, and *cpxA* mutants) exhibited sensitivity to various antibiotics. The detailed explanation of the relevant mechanisms should be covered in the discussion section.

Response: A novel paragraph discussing this issue was included in the revised manuscript as follows: “Systematic analysis of the two-component systems associated with the envelope stress response showed that several two-component systems affect intrinsic resistance to various antibiotics. The $\Delta envZ$ mutant was sensitive to vancomycin, fosfomycin, D-cycloserine, streptomycin, amikacin, tobramycin, and novobiocin, besides β -lactams (Fig. 1). Because OmpC and OmpF porins regulated by the EnvZ-OmpR two-component system act as a channel for the influx of various antibiotics and OmpC is also important for the maintenance of membrane integrity (17), these phenotypes of the $\Delta envZ$ mutant appears to be caused by altered expression levels of OmpC and OmpF. The $\Delta cpxA$ mutant was sensitive to chloramphenicol, erythromycin, nalidixic acid, tigecycline, and nitrofurantoin (Fig. 1). In *Klebsiella pneumoniae*, the sensitivity of the $\Delta cpxA$ mutant to chloramphenicol was reported (53), but its exact mechanism was not determined. Similarly, in *Haemophilus parasuis*, the sensitivity of the $\Delta cpxA$ mutant to erythromycin was reported (54), but its exact mechanism was not revealed. Notably, a recent study showed that CpxAR-mediated expressions of nitroreductases affect the prodrug activation of nitrofurantoin (55). Therefore, the sensitivity of the $\Delta cpxA$ mutant to nitrofurantoin could be caused by altered expression of nitroreductases. The $\Delta phoQ$ mutant was sensitive to cephalexin, erythromycin, and trimethoprim, besides tetracycline and tigecycline, whereas it was resistant to cloxacillin, cinoxacin, and novobiocin (Fig. 1). A recent study demonstrated that the PhoPQ system and MgrB affect susceptibility to trimethoprim by modulating the expression of dihydrofolate reductase, a target protein of trimethoprim (56). The relationship between the PhoPQ system and other antibiotics identified in this study has not been reported yet. Therefore, further studies are necessary to elucidate the molecular mechanisms underlying the resistance of the $\Delta phoQ$ mutant to these antibiotics.” (Lines, 310-331)

4. L282-285: Fig. 3B showed that the deletion of *eptB* significantly suppressed the sensitivity of the *phoP* mutant to bile salt. Like tetracycline and glycylcycline antibiotics, the bile salt also possesses a fused tetracyclic nucleus, which can be discussed in detail.

Response: Novel sentences discussing this issue were included in the revised manuscript as follows: “Notably, the deletion of *eptB* significantly suppressed the sensitivity of the $\Delta phoP$ mutant to bile salts (Fig. 5B). Like tetracycline and glycylcycline antibiotics, bile salts also possess a similar fused tetracyclic nucleus (40). Therefore, the relationship between the penetration of molecules with a fused tetracyclic nucleus across the outer membrane and KdoII modification can be investigated in further studies.” (Lines, 274-278)

II. Response to additional comments

- Introduction needs to be more concise.

Response: We abridged Introduction.

- Tetracyclines can't be considered as last resort antibiotics.

Response: You're right. We changed this part in the entire manuscript.

- do mutants defective in sensor kinases grow normally? Did you test for this?

Response: We added a novel figure (Fig. S1) showing the cell growth of mutant strains

in LB medium and added a novel sentence as follows: “All mutant strains did not show any growth defect in LB medium, except for slight growth retardation of the $\Delta cpx4$ mutant (Fig. S1).” (Lines, 97-98)

- Line 106: "various antibiotics", please specify the antibiotics.

Response: We specified the antibiotics.

- include the transposon mutagenesis library stats including the number of reads produced, coverage, saturation plots etc. Please include the stats in the text but the saturation plot should go to the supplementary material. As raised by the reviewer, this section also needs some work and clarification. In addition, sequencing reads should be submitted to GenBank and made publically available with the GenBank SRA numbers included in the revised manuscript.

Response: In our study, the transposon mutagenesis did not be performed using Tn-seq. The transposon mutagenesis was performed using the pir-dependent transposon delivery vector pRL27 carrying a Tn5 transposase gene and a mini-Tn5 element encoding kanamycin resistance. After selecting the mutant that suppresses doxycycline sensitivity of the $\Delta phoP$ mutant in an LB plate containing both kanamycin and doxycycline, the transposon insertion site was determined by PCR using primer sequences (an arbitrary primer consisting of a GGCGGT sequence and a random sequence, and a Tn5 transposon inner primer, 5'-GGTTGTAACACTGGCAGAGCATTACG-3'). PCR product was sequenced using another Tn5 transposon inner primer through Sanger sequencing. Therefore, we cannot include data sets of the transposon mutagenesis. Instead, we added a novel figure (Fig. S4) showing the precise position of the Tn5 insertion.

I appreciate you for the time in reviewing this submission.

With best wishes,

Prof. Chang-Ro Lee

Department of Biological Sciences
Myongji University
116 Myongjiro, Yongin
Gyeonggido, 449-728, Republic of Korea
TEL: +82-31-330-6472
FAX: +82-31-335-8249
e-mail: crlee@mju.ac.kr

Re: mSystems00964-24R1 (PhoPQ-mediated lipopolysaccharide modification governs intrinsic resistance to tetracycline and glycylicyline antibiotics in *Escherichia coli*)

Dear Prof. Chang-Ro Lee:

Your manuscript has been accepted, and I am forwarding it to the ASM production staff for publication. Your paper will first be checked to make sure all elements meet the technical requirements. ASM staff will contact you if anything needs to be revised before copyediting and production can begin. Otherwise, you will be notified when your proofs are ready to be viewed.

Sincerely,
Mehrad Hamidian
Editor
mSystems